# Curcumin as a Therapeutic Agent for Sarcopenia

**DOI:** 10.3390/nu15112526

**Published:** 2023-05-29

**Authors:** Siti Liyana Saud Gany, Kok-Yong Chin, Jen Kit Tan, Amilia Aminuddin, Suzana Makpol

**Affiliations:** 1Department of Biochemistry, Faculty of Medicine, Universiti Kebangsaan Malaysia, Kuala Lumpur 56000, Malaysia; 2Department of Pharmacology, Faculty of Medicine, Universiti Kebangsaan Malaysia, Kuala Lumpur 56000, Malaysia; 3Department of Physiology, Faculty of Medicine, Universiti Kebangsaan Malaysia, Kuala Lumpur 56000, Malaysia

**Keywords:** curcumin, muscle health, skeletal muscle, sarcopenia, ageing, therapeutic agent, oxidative stress, low-grade inflammation

## Abstract

Sarcopenia is the progressive loss of muscle mass, strength, and functions as we age. The pathogenesis of sarcopenia is underlined by oxidative stress and inflammation. As such, it is reasonable to suggest that a natural compound with both antioxidant and anti-inflammatory activities could prevent sarcopenia. Curcumin, a natural compound derived from turmeric with both properties, could benefit muscle health. This review aims to summarise the therapeutic effects of curcumin on cellular, animal, and human studies. The available evidence found in the literature showed that curcumin prevents muscle degeneration by upregulating the expression of genes related to protein synthesis and suppressing genes related to muscle degradation. It also protects muscle health by maintaining satellite cell number and function, protecting the mitochondrial function of muscle cells, and suppressing inflammation and oxidative stress. However, it is noted that most studies are preclinical. Evidence from randomised control trials in humans is lacking. In conclusion, curcumin has the potential to be utilised to manage muscle wasting and injury, pending more evidence from carefully planned human clinical trials.

## 1. Introduction

One in six individuals worldwide will be 60 years or older by 2030. The number of individuals 60 years and older worldwide is expected to increase from 1 billion in 2020 to 2.1 billion in 2050. By 2050, 426 million people will be 80 years or older [1]. Sarcopenia is often considered an important factor in senior frailty and mobility loss. It is characterised by decreased muscle mass, strength, and functionality with age [2,3,4,5,6,7,8]. An estimation in 2019 suggested that the economic burden of sarcopenia-associated disability was USD 40.4 billion in the United States, with an average cost of USD 260 per person. These expenses include hospitalisation, nursing home admissions, and home healthcare costs [9]. In addition to disability, sarcopenia is associated with multiple comorbidities, such as osteoporosis [10], obesity [11], and type II diabetes mellitus [12].

To effectively prevent sarcopenia, it is essential to have a thorough understanding of its pathophysiology and to use this knowledge to develop appropriate treatment strategies. The pathogenesis of sarcopenia includes impaired myofiber metabolism and adverse changes to muscle satellite cells, which lead to defective myogenesis and the loss of skeletal muscle homeostasis [13]. The reductions in muscle quality and strength associated with sarcopenia are also accompanied by neurological impairments affecting motor neurons and neuromuscular junctions, which result in denervated muscle fibers [14]. Additionally, persistent low-grade inflammation, poor anabolic signaling mediated by the growth hormone (GH)/insulin growth factor-1 (IGF-1) pathway, lower protein consumption, and vitamin D deficiency all contribute to the deterioration of muscle quality with age [15,16,17].

Skeletal muscle, the body’s primary metabolic organ, requires a lot of oxygen, micronutrients, and macronutrients to generate ATP for contraction [18,19]. In fact, during vigorous exercise, skeletal muscle tissue consumes 60% of the body’s total oxygen intake. On the other hand, skeletal muscle cells continuously break down glycogen and phosphocreatine to guarantee that anaerobic energy is available [18]. Skeletal muscle ageing is related to tissue metabolism reprogramming, affecting glucose, fat, and protein use and ultimately energy production [20]. Men lose more skeletal muscle and experience an increased visceral adipose with age, while women have less capillarisation of type II glycolytic myofibers [21]. The composition of the muscle fibres has an impact on the metabolism of macronutrients in aged skeletal muscle. In fact, type II fast-twitch fibres preferentially metabolise glucose anaerobically, while type I slow-twitch fibres preferentially metabolise fatty acids and are characterised by oxidation [22]. In the ageing skeletal muscle, decreased capillarisation and decreased nutrition transport to muscle cells have been observed [23,24]. 

With age, aerobic capacity, which is the maximum capacity to use oxygen to meet energy needs during both rest and exercise, tends to decline. In correspondence, their skeletal muscle energy metabolism decreases [25]. The mitochondria are crucial for the bioenergetic efficiency of skeletal muscle, and mitochondrial failure is widely acknowledged as a key indicator of ageing [26]. Age-related decrease in skeletal muscle mass and strength is caused by mitochondrial dysfunction. Maintaining mitochondrial health is essential for maintaining proteostasis in skeletal muscle. So far, studies on both animals and humans have contributed to a growing body of information regarding mitochondrial dysfunction in sarcopenia [27,28,29,30,31]. Both ATP depletion and ROS/RNS excess are linked to dysfunctional mitochondria, which cause the activation of dangerous cellular pathways. Aged skeletal muscle tissue exhibits a reduction in mitochondrial bulk, tricarboxylic acid cycle enzyme activity, oxygen consumption, and ATP generation [32]. Furthermore, apoptosis is triggered by mitochondrial dysfunction, which may degrade the quality of skeletal muscle [33].

Persistent low-grade inflammation has been recently demonstrated to impact both muscle protein synthesis and breakdown through several signaling pathways, which has an impact on sarcopenia [34]. Low-grade inflammation is a symptom of cells entering the senescent phase of the cell cycle and exiting the cell cycle. The pro-inflammatory tumour necrosis factor (TNF), IL-6, and C-reactive protein (CRP) are modestly elevated in the circulation in the course of ageing [34]. In conjunction, elderly people with sarcopenia are reported to have significantly higher levels of circulating IL-6 and TNF-α [35]. It was further revealed that elevated IL-6 and CRP levels increased the risk of losing muscle strength [36]. In addition, the plasma concentrations of TNF-α, IL-6, and IL-1 were reliable indicators of morbidity and mortality in senior participants in a 10-year longitudinal research study [37]. 

The conventional approach to sarcopenia management revolves around lifestyle changes, particularly through physical exercise and nutrition [38]. Due to growing awareness of the potential health benefits of natural products, the use of nutraceuticals, dietary supplements, and functional foods for disease prevention has been increasingly popular worldwide in recent years [39]. Given the importance of oxidative stress and inflammation in the aetiology of sarcopenia, compounds with antioxidant and anti-inflammatory characteristics have the potential to complement the current approach in treating this condition. Curcumin is one of the potential compounds with these qualities [40]. The turmeric-derived polyphenol curcumin (diferuloylmethane) has a variety of therapeutic uses. Since ancient times, turmeric has been used to relieve inflammation in traditional Chinese and Ayurvedic medicine [41]. According to recent scientific investigation, curcumin is reported to have antioxidant, anti-inflammatory, antimutagenic, antimicrobial, and anticancer properties [42,43,44,45,46]. Additionally, the precise molecular mechanisms of curcumin’s activity have been demonstrated through scientific studies [47]. Curcumin is demonstrated to target a variety of signaling molecules that modify cellular functions and perform its therapeutic effects [48]. 

This review aims to summarise the evidence of curcumin as a therapeutic agent to prevent sarcopenia. The biological effects of curcumin at the cellular level as well as in vivo are discussed. We hope this review will encourage interested researchers to investigate the use of curcumin as an interventional agent for sarcopenia and serve as a springboard for clinical studies on humans.

## 2. Therapeutic Effects of Curcumin on Muscle Health

Curcumin has been linked to a plethora of health advantages, including muscle health, in various studies [49]. Preserving muscle mass during ageing is the most important step in preventing sarcopenia. Receno et al. (2019) reported that curcumin (0.2% diet for 4 months) increased the muscular mass of supplemented rats without altering the body mass of 32-month-old male F344xBN rats, which is in line with previous studies [50,51,52,53]. In another study on contusion-induced muscular injury, 5 mg/kg body weight of curcumin for 7 days in 8-week-old male ICR mice was shown to dramatically enhance muscle mass. Similarly, curcumin supplementation boosted muscle mass but it had no discernible effect on body weight [54]. The higher skeletal muscle mass of curcumin-supplemented rats was consistent with earlier studies using various models of muscle wasting, by reducing both oxidative stress and inflammation [55,56].

The animals showed improved strength as a result of increased muscle strength. In a study, treatment of 150 mg/kg curcumin for over a period of 2 months resulted in improved muscle endurance, grip strength, and fat/lean mass ratio in 12-month-old male Sprague Dawley rats with LPS-induced sarcopenia [57]. Curcumin (40 and 80 mg/kg) administered 30 min before forced exercise for 28 days could complement exercise-based therapy to prevent muscular issues such as sarcopenia by regulating the expression of genes related to protein synthesis, apoptosis, and inflammation in chronic forced exercise executed 10-month-old ICR mice [58]. The effects of curcumin on muscle health are summarised in Table 1.

## 3. Mechanisms of Curcumin in Preserving Muscle Health

The beneficial effects of curcumin on muscle health are based on several mechanisms. Sani et al. (2021) investigated the protective effects of curcumin on dexamethasone-induced muscle atrophy using differentiated C2C12 cells. It was reported that curcumin treatment decreased Atrogin-1 and MuRF-1 expression, which inhibits protein degradation. It also increased the phosphorylation level of Akt, which is a vital protein in the mTOR signaling pathway which activates protein synthesis and inhibits protein degradation [59]. Gorza et al. (2021) demonstrated that curcumin, given to 18-month-old C57BL6J and C57BL10ScSN male mice for 6 months at a dose of 120 µg/kg, significantly increases satellite cell commitment and recruitment to delay the onset of pre-sarcopenia and sarcopenia. This was evident by the increased proportion of isolated MyoD-positive satellite cells from aged hindlimb muscles and sustained myofiber development in the ageing soleus muscle [49]. These mechanisms will be elaborated further in the following sections.

### 3.1. Effects of Curcumin on Satellite Cells

Sarcopenia is speculated to develop due to satellite cell malfunction and depletion [60]. Sarcopenia affects the ability of the elderly to restore muscular function by reducing the functionality of satellite cells [61]. Satellite cell loss reduces the ability of sarcopenic muscles to recuperate in old age and contributes to muscular fibrosis in response to injury [62]. Sarcopenia affects satellite function and maintenance, and throughout a person’s lifetime, reduces the ability of the muscle to regenerate new tissue and maintain tissue homeostasis [60]. 

Chronic inflammation harms satellite cells’ function in muscle regeneration [63]. This observation may be caused by the activation of the nuclear factor-κB (NF-κB) pathway in satellite cells [64]. This observation may be caused by the activation of the nuclear factor-κB (NF-κB) pathway in satellite cells [65], and inhibits the NF-κB pathway, thereby attenuating muscle protein degradation in other disease models such as sepsis [66,67]. 

In a recent study, treating immobilised 10-week-old C57BL/6J mice with 1 mg/kg body weight curcumin for 7 days demonstrated a significant improvement in the numbers of muscle progenitor cells, quiescent, activated, and total satellite cell counts in the limb muscles compared to non-treated immobilised animals [68]. In agreement with this, it has previously been demonstrated that upregulation of sirtuin-1 maintains satellite cells in a condition resembling stem cells. The increase in the quantity of quiescent and total satellite cells observed in the muscles of the unloaded animals treated with curcumin was most likely caused by this effect [69]. These results collectively imply that curcumin favors muscle regeneration after unloading. The effects of curcumin on satellite cells of the muscle are shown in Table 2. 

### 3.2. Effects of Curcumin on Mitochondrial Function

Mitochondria support energy equilibrium of the skeletal muscle through adaptive reprogramming in response to demands imposed by a variety of physiologic or pathological situations. Hamidie et al. (2015) demonstrated that the combination of curcumin therapy and endurance training could increase mitochondrial biogenesis in skeletal muscle via increasing cAMP levels [71]. In a different study, curcumin inhibited the activity of glycogen synthase kinase-3 (GSK-3) and was demonstrated to be useful in preventing muscular atrophy and mitochondrial dysfunction caused by chronic kidney disease (CKD). In the muscle of 6–7-week-old male C57BL/6 mice with CKD, GSK-3 deletion promoted mitochondrial biogenesis, decreased mitochondrial oxidative damage, and enhanced mitochondrial function. This study found that a diet containing 0.04% (*w*/*w*) curcumin treatment for 12 weeks prevented skeletal muscle GSK-3 activity, reducing the oxidative damage and dysfunction brought on by CKD in the mitochondria [72]. Another study using a mouse myoblast C2C12 cell line treated with low-dose (1, 2 and 5 uM) and high-dose (10, 20 and 50 uM) curcumin, discovered that curcumin might be an effective treatment for diabetes and other mitochondrial diseases when used in modest dosages (5 uM) [73]. In a separate study, it was demonstrated that curcumin supplementation of 100 mg/kg body weight for 3 days significantly increased the activity of mitochondrial enzymes cytochrome c oxidase, succinate dehydrogenase, Na+/K+-ATPase, and Ca2+-ATPase in the skeletal muscle mitochondria of rats with chronic obstructive pulmonary disease (COPD) [74]. The effects of curcumin on the mitochondrial function of the muscle are shown in Table 3.

### 3.3. Effects of Curcumin on Low-Grade Inflammation

Evidence has shown that chronic low-grade inflammation led to sarcopenia because it influences both muscle protein production and breakdown through several signaling pathways [34]. Low-grade inflammation is, at least in part, a result of an increase in the number of cells that exit the cell cycle and enter cellular senescence [34]. A previous study on C2C12 cells treated with 100 ug/mL Cur-SHAP showed decreased levels of IL-6 and TNF-α, suggesting that curcumin may inhibit inflammation induced by LPS [57].

In another study, 80 women with moderate physical levels given oral curcumin supplementation of 500 mg/day for 8 weeks showed a considerable drop in CRP and lactate dehydrogenase (LDH) levels, which is a marker for muscle and tissue damage. Additionally, curcumin administration also reduced MDA significantly, which had a favorable impact on oxidative stress [75]. This shows that there are very limited studies on the anti-inflammatory effect of curcumin on skeletal muscle health, and more research should be conducted to learn more about it. The effects of curcumin on inflammation and muscle health in humans are shown in Table 4.

### 3.4. Effects of Curcumin on Oxidative Stress

The redox status of our bodies is regulated by various antioxidant enzymes that break down potential oxidants. Superoxide dismutase (SOD) converts superoxide into H_2_O_2_ and O_2_ [76]. Catalase produces non-toxic H_2_O and O_2_, with H_2_O_2_ produced by SOD [77]. Additionally, glutathione peroxidase (GPx) transforms reduced glutathione and H_2_O_2_ into glutathione disulfide and H_2_O [78]. Unremoved ROS react with polyunsaturated fatty acids to form lipid peroxidation, which results in MDA production [79]. A rise or fall in the ROS level can be verified by monitoring the MDA level and antioxidant enzyme activity.

Increased oxidative stress in skeletal muscle is a hallmark of ageing, which can disrupt cellular redox control, alter transcription factor activity, and damage cellular macromolecules such as proteins, lipids, and DNA [80]. Age-related changes in oxidative load may be brought about by increasing oxidation, diminished antioxidant defenses, or a combination of these factors [80]. For instance, aged skeletal muscle produced more ROS and reactive nitrogen species, whereas the antioxidant state of the human vastus lateralis decreased, indicating a degraded functional response [81]. A study on dexamethasone-induced (DEX) sarcopenia reported that DEX increases ROS to produce muscle atrophy [82].

Curcumin intake has been shown to increase antioxidant enzyme activities and decrease the levels of MDA [83]. A prior study using C2C12 cells treated with 1 ug/mL lipopolysaccharides (LPS) to stimulate inflammation and then treated with 100 ug/mL curcumin stearic acid (Cur-SHAP) after 24 h showed that ROS levels in the Cur-SHAP group were lower than that in the LPS group, which confirmed that Cur-SHAP exhibits good antioxidant effects [57]. In a separate study, C2C12 myoblasts were treated with turmeric hot water extract at 100, 250, 500, 750, and 1000 ug/mL concentrations for 24 h [84]. The study reported that curcumin treatment reduced intracellular ROS levels in the H_2_O_2_-induced cells, suggesting that curcumin might have the ability to protect oxidative stress-induced C2C12 myoblasts [84]. 

Kim S. et al. conducted a study of dexamethasone-induced sarcopenia in ICR mice fed with curcumin water extract for 1 week, which reported that myostatin, MuRF-1, and Atrogin-1 expression levels were decreased, thus preventing muscle loss. Additionally, the study also reported that curcumin boosted the antioxidant enzyme activity while lowering MDA levels [83]. The transcription factor nuclear factor erythroid-2 related-factor-2 (Nrf2) is the “master transcriptional regulator” of the body’s antioxidant defenses [85]. When the production of ROS is enhanced, Nrf2 binds to the antioxidant response element (ARE) and increases the expression of the genes for antioxidant enzymes [86]. Numerous studies have linked the dysregulation of Nrf2 with ageing and enhanced oxidative stress, decreased antioxidant response, and muscle deterioration. Therefore, activating Nrf2 and reducing oxidative stress would benefit the ageing skeletal muscle [87,88,89]. Curcumin can neutralise free radicals directly, and encourage nuclear translocation and Nrf2 activation via separation from the kelch-like ECH-associated protein 1 (Keap1) [90]. He et al. (2012) found that after 15 days of 50 mg/kg body weight curcumin treatment in a mouse model of high-fat diet-induced insulin resistance, muscular Nrf2 activation was greater compared to untreated controls [91]. In another study, 32-month-old F344xBN male rats given a 0.2% curcumin-supplemented diet for 4 months showed that curcumin increases levels of Nrf2. Additionally, curcumin also significantly lowered 3-nitrotyrosine (3-NT) and protein carbonyls (PC) levels [53]. The evidence of curcumin improving muscle health through oxidative stress regulation is shown in Table 5. The mechanism effect of curcumin is summarised in Figure 1.

## 4. Conclusions

There are no effective pharmaceutical treatments for sarcopenia at this moment [93]. The existing evidence shows that curcumin can be a potential alternative treatment for managing sarcopenia. It achieves muscle protection by maintaining satellite cell number and function, protecting the mitochondrial function of muscle cells, as well as suppressing inflammation and oxidative stress. However, more studies should be conducted on the delivery route, exact dose, and mechanisms of action of curcumin on sarcopenia. Its muscle protective functions should be verified in well-planned human clinical trials to ensure efficacy and safety in humans.

## Figures and Tables

**Figure 1 nutrients-15-02526-f001:**
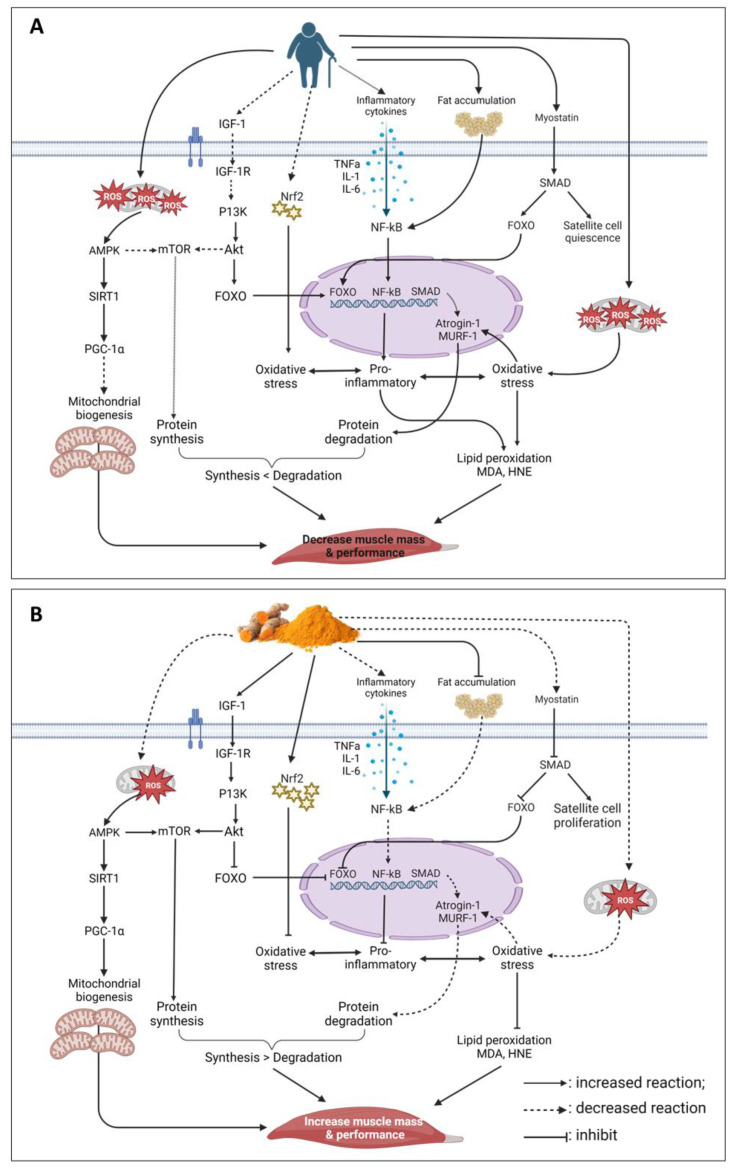
Schematic diagram illustrating the effects of ageing on muscle mass and performance. (**A**) Mechanism of curcumin in maintaining muscle mass and performance (**B**). Regulation mechanisms that contribute to the protective effects of curcumin in ageing skeletal muscle: (1) inhibition of intracellular ROS formation and decreased oxidative stress markers; (2) inhibition of intracellular ROS formation and increased mitochondrial biogenesis; (3) inhibition of fat accumulation and reduced inflammatory cytokines. (
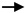
: increased reaction; 
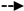
: decreased reaction; 
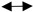
: reversible reaction; 
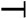
: inhibit).

**Table 1 nutrients-15-02526-t001:** Effects of curcumin on muscle health sarcopenia.

Author (Years)	Study Design	Major Findings	Conclusion
Lee et al., 2021 [58]	80 and 40 mg/kg curcumin administered 30 min before forced exercise for 28 days on 10-month-old male Hsd/ICR (CD-1) mice.	↑ calf thicknesses and strengths, total body and calf protein amounts, and muscle weights in gastrocnemius and soleus muscles ↓ MDA levels and ROS contents in gastrocnemius and soleus muscle ↑ GSH contents, SOD and CAT in gastrocnemius and soleus muscle ↓ mRNA expressions in gastrocnemius and soleus muscle ↑ protein synthesis ↑muscle hypertrophic changes ↑ ATPase-immunoreactive fibers and normalised myostatin-immunoreactive fibers ↓ muscular caspase-3 and PARP immunoreactivities ↓ nitro tyrosine, 4HNE and iNOS-specific muscle fibers.	Curcumin can improve muscle health in combination with exercise. Appropriate dosages of curcumin for muscle health improvement require further investigation in both animals and humans.
Liang et al., 2021 [57]	Sprague Dawley rats with muscle injury given 150 mg/kg curcumin (Cur-SHAP) 4 times over 42 days before being injected with LPS twice a week. 200 mg Curcumin + 200 mL ddH_2_O and mixed with 500 mg of SHAP particles were used in this study.	↑ muscle endurance as per 30-min treadmill test Fully recover grip strength ↓ TP, CL, LDH, calcium and ALT values compared to LPS group	Hydroxyapatite and hydrophobic surface modification are used to load curcumin for intramuscular drug administration. This method helped rats with LPS-induced sarcopenia to regain their health.
Tsai et al., 2020 [54]	8-week-old male ICR mice given oral doses of curcumin at 10 mg/kg and 5 mg/kg BW once daily for 7 days after injury.	↑food intake in muscle injured rats ↑ muscle mass ↑ functional recovery of gait speed test ↓ serum uric acid, CK levels, MDA levels ↓ protein levels of Ikk-α/β, MPO, CD206 and myogenin ↓ disruption of muscle tissues in contusion-induced muscle injury	Curcumin may affect inflammation, neutrophil, and satellite cell differentiation proteins. Curcumin has potential to speed up muscle healing. Further research is needed to determine curcumin usefulness in medicine for muscle restoration.
Sani et al., 2021 [59]	Different amounts of curcuminoids were grown and given to C2C12 cells (in vitro model of muscle atrophy).	↓ Atrogin-1 gene expression ↓ MuRF-1 gene expression ↑ p-AKT	Curcumin can prevent muscle atrophy. These effects make them promising therapeutic agents for treating sarcopenic patients.
Gorza et al., 2021 [49]	Male C57BL6J and C57BL10ScSn mice aged 18 months treated with curcumin for 6 months. 100 uL volume of 50 mg/mL curcumin in ethanol added to 1.5 mL of 100 mg/mL HPβCD in 0.15 M NaCl, 0.2 mM PBS pH 7.4 was used in this study.	↑ specific tetanic tension ↑ pure type 2x myofibers ↑ EDL myofibers ↓ dystrophin protein levels and further melusin amounts. ↑ SERCA1 protein levels in old soleus ↓ SERCA 1 protein levels in old EDL ↓ Tas in EDL muscles	Curcumin can reverse some age-related changes in muscle proteins and distribution of costamere components. Helps maintain adult muscle levels in regeneration muscles. Increased the number of satellite cells in aged muscles. Delays onset of muscle loss.
Receno et al., 2019 [53]	Male F344xBN rats aged 32 months were fed either a diet containing 0.2% curcumin or a control diet for 4 months.	↓ Food intake ↑ Nrf2 levels in plantaris muscle ↓ 3-NT and PC levels ↑ specific peak twitch and specific tetanic tension response	Curcumin consumption along with less food intake has positive effects on skeletal muscle in older people. Variability in measurements may be due to the absence of curcumin at tissue or systemic levels.
Liu et al., 2016 [56]	Male C57BL/6J mice aged 10–12 weeks treated with 100 mg/kg curcumin solution via IP 1 hour before ligation surgery.	↑ running capacity ↑ muscle regeneration (↑ fiber density and ↓ fibrosis) ↓ macrophage infiltration in ischemic muscle ↓ TNF-α, IL-1β and IL-6 in limb muscle tissue ↓ levels of NF-κB-p65	Curcumin can inhibit activation of NF-κB in macrophages induced by LPS. Can help improve hindlimb injury after ischemic surgery, indicating its potential use for treating PAD.
Ono et al., 2015 [55]	C57BL/6J 8–10 weeks mice fed with 1500 mg/kg curcumin for 2 weeks and compared to DM group.	↓ body weight Improve myocyte cross-sectional area ↓ ubiquitin-conjugated proteins. ↓ atrogin-1/MAFbX and MuRF-1 ↓ TNF-α and IL-1β ↓ superoxide and TBARS	Curcumin could be a useful treatment for muscle atrophy in type 1 DM. Antioxidant properties of curcumin are not well-defined and more research is needed.

Abbreviations: ROS: reactive oxygen species; ATPase: adenosine triphosphate; IP: intraperitoneal; IM: intramuscular; LPS: lipopolysaccharide; IKK α/β: IkappaB kinase; MPO: myeloperoxidase; TNF-a: tumor necrosis alpha; IL-6: interleukin 6; PAD: peripheral arterial disease; DM: diabetes mellitus; MuRF-1: muscle ring finger; p-AKT: phosphorylation of AKT; EDL: extensor digitorum longus; TA: tubular aggregates; SERCA1: fast skeletal calcium-pump; Nrf2: nuclear factor erythroid-2 related-factor-2; 3-NT: 3-nitrotyrosine; PC: protein carbonyls; PAD: peripheral artery disease; TBARS: thibarbituric acid reactive substances; Cur-SHAP: curcumin-loaded HAP (hydroxyapatite); TP: total protein; CK: creatine kinase; LDH: lactate dehydrogenase; Ca: calcium; ALT: alanine aminotransferase; HPβCD: hydroxypropyl-β-cyclodextrins; PBS: phosphate buffer. (↑: increase; ↓: decrease).

**Table 2 nutrients-15-02526-t002:** The effects of curcumin on muscle cells.

Author (Years)	Study Design	Major Findings	Conclusion
Grabowska et al., 2016 [65]	Cells senescing in a replicative and premature manner were exposed to curcumin at low doses (0.1 and 1 µM)	↑ sirtuin 1 and 6 ↑ number of cells with elevated activity of SA-β-gal ↓ IL-8 and VEGF ↑ levels and activity of AMPK and phosphorylation of ACC	Sirtuin activation could be caused by the activation of AMPK brought on by ↑ in superoxide and ↓ ATP. ↑ level of sirtuins without delaying the VSMC’s senescence.
Liang et al, 2021 [57]	C2C12 myoblasts were cultured in a 24-well plate at a density of 3 × 10^4^ cells/well and incubated for 24 h. 0.1 mg/mL HAP and Cur-SHAP added into each well and incubated for 24 h LIVE/DEAD staining was observed using a fluorescence microscope	↓ ROS levels ↓ IL-6, TNF-α and Atrogin-1	Cur-SHAP confirmed to exhibit good antioxidant effect Cur-SHAP decrease gene expression level of IL-6 and TNF-α to inhibit LPS-induced inflammation
Jin and Li, 2007 [66]	Daily IP injection of 10–60 ug/kg of curcumin for 4 days in LPS-induced muscle wasting in adult male ICR mice	↓ muscle protein loss. Inhibits LPS stimulation of atrogin-1/MAFbx expression. Inhibits LPS-induced activation of p38-MAPK in muscle.	Curcumin can prevent loss of muscle mass induced by LPS by inhibiting p38-MAPK/MAFbx Unknown why curcumin does not have previously described inhibitory effect on NF-κB activation
Poylin et al., 2008 [67]	Male Sprague-Dawley rats were induced with sepsis by cecal ligation and puncture (CLP) or were sham-operated, treated with IP doses of curcumin (600 mg/kg) or equivalent amounts of solvent	↓ sepsis-induced muscle proteolysis ↓ proteasome-, calpain- and cathepsin L-dependent protein breakdown rats ↓ catabolic response to sepsis by inhibiting multiple proteolytic pathways. ↓ NF-κB/p65 expression and activity levels and phosphorylated (activated) p38 were ↓	Curcumin prevents muscle breakdown in sepsis Does not affect expression of atrogin-1 and MuRF1. Muscle protein breakdown rates do not correlate with changes in atrogin-1 and MuRF1 expression during muscle wasting treatment.
Thaloor et al., 1999 [70]	4–6-week-old C57BL/6 male mice subject to a standardised freeze injury on Masseter or tibialis anterior (TA) muscles. Mice injected IP with 0.15–0.2 mL of either curcumin or vehicle (DMSO) diluted in PBS starting on day of the damage and continuing once daily thereafter.	↑ EMHC expression ↑ largely nucleated myofibers (muscle regeneration) ↑ fusion of myoblasts Inhibit NF-κB mediated transcription in myoblasts.	Curcumin improves muscle regeneration in multiple muscles in same animal. The effect of curcumin is dose-dependent and occurs soon after damage.
Manas-Garcia et al., 2020 [68]	Female C57BL/6J exposed to recovery after 7-day hindlimb immobilisation treated with 1 mg/kg/bw/day for 7 days.	↓ TUNEL-positive nuclei ↑ size of hybrid fibers ↑ sirtuin-1 activity in gastrocnemius muscle ↓ troponin-1 levels ↓ muscle atrogin-1, MuRF-1 and total protein ubiquitination ↑ levels of puromycin-labeled proteins and phosphorylated Akt. ↓ total NF-κB p50 subunit and Fox01 in gastrocnemius muscle ↑ protein levels of HDAC4	Earlier findings may differ from current study due to changes in experimental models, animal strains, protocol duration, and type of study muscles. Curcumin reduces muscle proteolysis by activating Sirtuin-1, thus decreasing activity of atrophy signaling pathways.

Abbreviations: IP: intraperitoneal; AMPK: activated protein kinase; ATP: adenosine triphosphate; VSMC: vascular smooth muscle cells; VEGF: vascular endothelial growth factor; EDL: extensor digitorum longus; LPS: lipopolysaccharide; CLP: cecal ligation and puncture; DMSO: dimethyl sulfoxide; EHMC: embryonic myosin heavy chain; TA: tibialis anterior; PGC-1α: peroxisome proliferator-activated receptor-γ coactivator; HDAC: histone deacetylases. (↑: increase; ↓: decrease).

**Table 3 nutrients-15-02526-t003:** The effects of curcumin on mitochondrial function of the muscle.

Author (Years)	Study Design	Major Findings	Conclusion
Ray Hamidie et al., 2015 [71]	10-week-old male Wistar rats undergo eTR given high dose (50 mg/kg-BW/day) or high dose (100 mg/kg-BW/day) of curcumin dissolved in DMSO injected IP for 28 days.	↓ body weight ↑ COX-IV and OXPHOS subunits in gastrocnemius muscle ↑ mtDNA copy number and ↑ CS activity in gastrocnemius muscle ↑ AMPK ↑ SIRT1 and NAD+/NADH ratio ↓ acetylation of PGC1-α ↑ cAMP, phosphorylation of CREB and LKB-1	Curcumin + eTR can potentially increase cAMP levels and speed up mitochondrial biogenesis in skeletal muscle.
Wang et al., 2020 [72]	6–7-week-old male C57BL/6 CKD mice fed diet containing 0.04% curcumin for 12 weeks.	↓ Body weight ↑ lean muscle mass, gastrocnemius, and soleus muscle ↑ grip strength and running distance ↓ mitochondrial O_2_^−^ ↑ mitochondrial ATP and OCR ↓ disruption of mitochondrial ETC enzyme activity ↑ mTFAM, PGC-1α and NRF-1	Curcumin ↓ CKD-related mitochondrial oxidative damage and dysfunction through preventing GSK-3 activity in skeletal muscle.
Yu et al., 2019 [73]	Mouse myoblast C2C12 cell line treated with 6 different curcumin in DMSO concentration at low dose (1, 2 and 5 µM) and high dose (10, 20 and 50 µM) for 24 h.	↑ mitochondrial mass at low concentration (5 µM). ↑ cell viability at low dose	New or alternate formulations should be developed to manipulate curcumin concentration within cells.
Zhang et al., 2017 [74]	COPD male Sprague Dawley rats treated with 100 mg/kg body weight curcumin dissolved in 0.1% carboxy methylcellulose-Na solution for 3 days.	↑ body weight ↑ COX, SDH, Na^+^/K^+^-ATPase and Ca^2+^-ATPase ↓ MDA ↑ MnSOD, GSH-Px and catalase ↓ IL-6 and TNF-α ↑ PGC-1α and SIRT3	Curcumin can reduce the mitochondrial dysfunction in the skeletal muscle of COPD rats, probably by enhancing the PGC-1/SIRT3 signaling pathway.

Abbreviations: eTR: endurance training; BW: body weight; IP: intraperitoneal; AMPK: activated protein kinase; NAD: nicotinamide adenine dinucleotide; NADH: nicotinamide adenine dinucleotide + hydrogen; Sirt1: silent information regulator 1; PGC-1α: peroxisome proliferator-activated receptor-γ coactivator; cAMP: cyclic adenosine monophosphate; mtDNA: mitochondria DNA; COX-IV: cytochrome c oxidase subunit IV; OXPHOS: oxidative phosphorylation; CS activity: citrate synthase activity; CREB: cAMP response element binding protein; LKB-1: liver kinase B-1; CKD: chronic kidney disease; ATP: adenosine triphosphate; ROS: reactive oxygen species; COPD: chronic obstructive pulmonary disease; ETC: electron transport chain; TFAM: mitochondrial transcription factor A; NRF-1: nuclear response factor-1; SOD: superoxide dismutase; GSH: glutathione; GPx: glutathione peroxidase; GR: glutathione reductase; GSK-3β: glycogen synthase kinase-3β; COX: cytochrome c oxidase; SDH: succinate dehydrogenase; MDA: malondialdehyde; MnSOD: manganese superoxide dismutase; GSH-Px: glutathionie peroxidase. (↑: increase; ↓: decrease).

**Table 4 nutrients-15-02526-t004:** The effects of curcumin on inflammation and muscle health.

Author (Years)	Study Design	Major Findings	Conclusion
Liang et al., 2021 [57]	Sprague Dawley rats with muscle injury given 150 mg/kg curcumin four times over 42 days before being injected with LPS twice a week.	↓ IL-6, TNF-α and Atrogin 1 ↑ muscle endurance as per 30-min treadmill test Fully recover grip strength	Cur-SHAP can decrease the gene expression level of IL-6 and TNF-α to inhibit inflammation induced by LPS.
Salehi et al., 2021 [75]	Double-blind, placebo-controlled clinical trial on 80 women with moderate physical activity levels given either 500 mg/day curcumin for 8 weeks. 10 cc of blood was obtained at the beginning and end of the study.	↓ CRP, LDH. MDA levels ↑ VO2 max	Curcumin’s impact on women’s muscle health was studied, but not on men or professional athletes. Gene expression and molecular mechanisms were not investigated. Future research should examine gene markers and the effects of various curcumin doses.

Abbreviations: CRP: C-reactive protein; LDH: lactate dehydrogenase; MDA: malondialdehyde; VO2 max: maximal oxygen consumption; FFQ: food frequencies questionnaire; CM-SD: curcumin spray dry; LPO: lipid peroxidation; ROS: reactive oxygen species. (↑: increase; ↓: decrease).

**Table 5 nutrients-15-02526-t005:** The effects of curcumin on muscle health through the regulation of oxidative stress.

Author (Years)	Study Design	Major Findings	Conclusion
Liang et al., 2021 [57]	Inflammed C2C12 cells treated with 100 ug/mL Cur-SHAP for 24 h	↓ ROS level	Cur-SHAP confirmed to exhibit good antioxidant effects.
Wang et al., 2020 [72]	6–7 week-old male C57BL/6 CKD mice fed a diet containing 0.04% curcumin for 12 weeks	↓ MDA in muscle tissue ↑ SOD, GSH, GPx and GR in quad muscle tissue ↓GSK-3β expression in vivo and in vitro ↓ NRF-1, MAFbx and chymotrypsin and trypsin-like activities	Curcumin ↓ CKD-related mitochondrial oxidative damage and dysfunction through preventing GSK-3 activity in skeletal muscle.
Yu et al., 2019 [73]	Mouse myoblast C2C12 cell line treated with 6 different curcumin in DMSO concentration at low dose (1, 2 and 5 µM) and high dose (10, 20 and 50 µM) for 24 h	↓ cellular ROS level at low dose	ROS equilibrium in cells responds according to different curcumin concentrations. New or alternate formulations should be developed to manipulate curcumin concentration within cells.
Jeong et al., 2017 [84]	Oxidative stress-induced C2C12 myoblasts treated with curcumin. (amount and duration were not mentioned)	↑ cell viability against oxidative stress-induced cell death ↓ intracellular ROS levels in cells	Curcumin can protect oxidative stress induced C2C12 myoblasts.
Kim et al., 2021 [83]	Dexamethasone-induced sarcopenia in ICR mice treated with curcumin powder dissolved into 20 folds water at 250 °C for 3 h for 1 week Blood samples were collected transcardially via the apex of the left ventricle and tissue samples were collected and stored at −80 °C	↓ expression levels of myostatin, MuRF-1 and Atrogin-1 ↑ antioxidant enzyme activity and ↓ MDA levels	Curcumin protects muscular atrophy by altering genes associated with it and boosting antioxidant capacity.
Sahin et al., 2016 [90]	Male 8-week-old Wistar rats were administered with 100 mg/kg curcumin daily for 6 weeks together with exercise	↓ Lactate and MDA in tissue muscle ↑ antioxidant activities in blood sample ↓ muscle NF-kB and heat shock protein levels in blood sample ↑ Nrf-2 and glucose transporter 4 protein levels	Curcumin can help prevent muscle damage by regulating the NF-kB and Nrf2 pathways.
He et al., 2012 [91]	Obese male C57BL/6J treated with 50 mg/kg/bw curcumin via oral gavage for 15 days	↓ glucose intolerance ↓ MDA and ROS in serum and muscle tissue respectively ↑ Nrf2 signaling	Short-term therapy of curcumin significantly reduces muscle oxidative stress in HFD-fed animals.
Receno et al., 2019 [53]	32 months old F344xBN male rats supplemented with 0.2% curcumin for 4 months	Curcumin ↑ Nrf2 compared to curcumin + normal diet rats in muscle tissue Levels of 3-NT and PC ↓ in curcumin group compared to curcumin + normal diet rats in muscle tissue	Consuming curcumin and reducing food intake together improved skeletal muscle health in aged skeletal muscle.
Franceschi et al., 2016 [92]	86 healthy subjects >65 years old supplemented with 1 g Meriva (curcumin) daily for 3 months.	↓ oxidative stress ↓ proteinuria	Meriva on its own or combined with other nutritional supplements can improve strength and physical performance in elderly subjects, potentially preventing the onset of sarcopenia.

Abbreviations: Cur: curcumin; SHAP: stearic acid; ROS: reactive oxygen species; LPS: lipopolysaccharides; MDA: malondialdehyde; NF-kB: nuclear factor kappa beta; Nrf2: nuclear factor erythroid 2-related factor 2; HFD: high-fat diet. PC: protein carbonyls. (↑: increase; ↓: decrease).

## Data Availability

Not applicable.

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
