# Peer review of "Curcumin as a Therapeutic Agent for Sarcopenia"

_nutrients, 2023, doi:10.3390/nu15112526_

Round 1

Reviewer 1 Report

There are some grammatical errors, such as, line 17, lines 28-30, line 113, line 121.

Tables 1-5, illustrate up arrow and down arrow. In column Major findings, arrows are used by mistake.

There are some grammatical errors, such as, line 17, lines 28-30, line 113, line 121.

Author Response

Thank you very much for your comments.  I have made the corrections based on your recommendations.

1. There are some grammatical errors, such as, line 17, line 28-30, line 113, and line 121.

Thank you. I have corrected the grammatical errors.

2. Tables 1-5, illustrate up arrow and down arrow. In column Major findings, arrows are used by mistake.

Thank you. I have corrected the arrows.

Reviewer 2 Report

This study summarized the effect of curcumin on sarcopenia. The contents of this review was interesting and comprehensive.

This reviewer has a few comments.

1. For each table, please summarize the target species or materials, materials and quantities used, and the major finding with target genes or pathways.

2. For Figure 1, please describe the meaning of arrows and dotted arrows.

Furthermore, the resolution of the figure is low and the ROS is illegible.

Author Response

Thank you very much for your comments. I have made the corrections based on your recommendations. 

  1. For each table, please summarize the target species or materials, materials and quantities used, and the major finding with target genes or pathways.

Thank you. I have summarized the table accordingly.

  1. For figure 1, please describe the meaning of arrows and dotted arrows.

Thank you. I have described the meaning of arrows and dotted arrows for figure 1.

  1. The resolution of the figure is low and the ROS is illegible.

Thank you. I have edited the figure to higher resolution.

Reviewer 3 Report

Dear authors,

Thank you for writing this review on Curcumin and its effects on sarcopenia. The manuscript is well-written and informative. I only have some minor edits and suggestion in the hope of improving the quality of this work which is listed below.

1.  Line 12 I would suggest adding "chronic" before oxidative stress. Oxidative stress is normal to some extent so this could be misleading.

2. Line 36 "It should be added that" remove this.

3. Line 39 Why only treatment? how about prevention? Please expand on this.

4. Line 48-49 "Skeletal muscle, being the body’s primary metabolic organ, requires a lot of oxygen 48 and macronutrients to generate ATP for contraction"  please add a reference for this statement. 

5. Line 48-49 I think both macro/micronutrients are required for ATP production.

6. Line 120 "With increased muscle strength, the strength of the animals was improved.Did you mean overall strength? Please consider re-wording this.

Line 121 "In a 120 study, treatment of 150 mg/kg curcumin for .... resulted in improved..."  Please add the duration of supplementation after "for".

Line 278 Table 5. Just ensure that the formatting is consistent throughout the manuscript. 

Minor editing of English language required.

Author Response

Thank you very much for your comments. I have made the corrections based on your recommendations. 

  1. Line 12 I would suggest adding “chronic” before oxidative stress. Oxidative stress in normal to some extent so this could be misleading.

Thank you. I have added “chronic” before oxidative stress in line 12.

  1. Line 36 “It should be added that” remove this.

Thank you. I have removed the said line.

  1. Line 39 Why only treatment? How about prevention? Please expand on this.

Thank you. I have expanded on this.

“To effectively prevent sarcopenia, it is essential to have a thorough understanding of tis pathophysiology and to use this knowledge to develop appropriate treatment strategies.”

  1. Line 48-50 “Skeletal muscle, being the body’s primary metabolic organ, requires a lot of oxygen 48 and macronutrients to generate ATP for contraction” please add a reference for this statement.

Thank you. I have added a reference for this statement (now line 50-51)

  1. Line 48-49 I think both macro/micronutrients are required for ATP production.

Thank you. You are right. I have included micronutrients in the sentence as well.

“Skeletal muscle, being the body’s primary metabolic organ, requires a lot of oxygen as well as micronutrients and macronutrients to generate ATP for contraction”

  1. Line 120 “With increased muscle strength, the strength of the animals was improved”. Did you mean overall strength. Please consider re-wording this.

Thank you. I have rephrased the sentence.

The animals showed improved strength as a result of increased muscle strength.

  1. Line 121 “In a 120 study, treatment of 150 mg/kg curcumin for…. Resulted in improved…” Please add the duration of supplementation after “for”

Thank you. I have added the period of supplementation which is 2 months.

  1. Line 278 Table 5. Just ensure that the formatting is consistent throughout the manuscript.

Thank you. I have reformatted the table accordingly.